# A novel open-source raspberry Pi-based behavioral testing in zebrafish

**Yunlin Li, Fengye Wu, Qinyan Wu, Wenya Liu, Guanghui Li, Benxing Yao, Ran Xiao, Yudie Hu, Junsong Wang** * 

Center for Molecular Metabolism, School of Environmental & Biological Engineering, Nanjing University of Science and Technology, Nanjing, China

* wang.junsong@gmail.com

**Data Availability Statement:** All relevant data are within the paper and its Supporting information files.

**Funding:** This research was funded by the National Natural Science Foundation of China (Grant No. 81773857).

## Abstract

The zebrafish (*Danio rerio*) is widely used as a promising high-throughput model organism in neurobehavioral research. The mobility of zebrafish can be dissected into multiple behavior endpoints to assess its neurobehavioral performance. However, such facilities on the market are expensive and clumsy to be used in laboratories. Here, we designed a low-cost, automatic zebrafish behavior assay apparatus, barely without unintentional human operational errors. The data acquisition part, composed of Raspberry Pi and HQ Camera, automatically performs video recording and data storage. Then, the data processing process is also on the Raspberry Pi. Water droplets and inner wall reflection of multi-well cell culture plates (used for placing zebrafish) will affect the accuracy of object recognition. And during the rapid movement of zebrafish, the probability of zebrafish tracking loss increased significantly. Thus, ROI region and related thresholds were set, and the Kalman filter algorithm was performed to estimate the best position of zebrafish in each frame. In addition, all functions of this device are realized by the custom-written behavior analysis algorithm, which makes the optimization of the setup more efficient. Furthermore, this setup was also used to analyze the behavioral changes of zebrafish under different concentrations of alcohol exposure to verify the reliability and accuracy. The alcohol exposure induced an inverted U-shape dose-dependent behavior change in zebrafish, which was consistent with previous studies, showcasing that the data obtained from the setup proposed in this study are accurate and reliable. Finally, the setup was comprehensively assessed by evaluating the accuracy of zebrafish detection (precision, recall, F-score), and predicting alcohol concentration by XGBoost. In conclusion, this study provides a simple, and low-cost package for the determination of multiple behavioral parameters of zebrafish with high accuracy, which could be easily adapted for various other fields.

## Introduction

Zebrafish is a common tropical freshwater fish, which has become a powerful model organism in biological research [1]. Compared with other experimental animals, zebrafish have the advantages of small size, high economic benefits, easy breeding, and great similarity (70%) to

**Competing interests:** The authors have declared that no competing interests exist.

human disease genes [2, 3]. Nowadays, zebrafish have been widely used in developmental, genetics, cardiovascular, and neurobiological research for its unique advantages [4, 5]. Therefore, zebrafish has become an alternative animal model for studying diseases and screening drugs on a large scale.

Behavioral experiments study the natural animal activities macroscopically, which directly reflect neurological performance of the body in response to external stimuli, and thus are indispensable to and complemented well with other microscopic analyses [6]. Compared with other conventional evaluation systems, such as histopathological and biochemical parameters, the assessment of behavior can more simply and directly reflect the body's physiological state to judge the overall therapeutic drug effects on diseases in some degree. Nowadays, animal behavior are usually observed by expensive commercial devices or human eyes, which are time- and labor-intensive, defying high-throughput detection. In addition, video capturing of animal behavior and subsequent computer analysis can automatically obtain and process a large number of video signals to give accurate description of behavior phenotype, allowing high-throughput study of animal behavior changes [7–9]. However, most of the zebrafish behavior assay setup on the market is sophisticated and expensive [10–13], thus greatly limited its application.

Raspberry Pi microprocessor has received extensive attention since its release in 2012 due to its powerful function, low price, and convenient operation. It has been widely used in laboratories, education, robots, and other fields [14–17]. In addition, Raspberry Pi's rich remote-operated features can greatly reduce experimental bias and error, thus easy for automation to obtain huge amount of accurate dataset [9]. These advantages make Raspberry Pi an appealing choice to develop low-cost, and high-throughput animal behavior assay setup.

In this study, an automatic video capture setup based on Raspberry Pi, and a zebrafish behavior assay method was established. The video acquisition module designed in this study minimized the influence of environmental factors such as the operators, temperature and sound, thereby reduced the unintentional human errors caused by unwanted non-experimental factors. In addition, by setting the region of interest (ROI) and performing the object recognition algorithm, the accuracy of object detection is improved, and the interference of water droplets and shadows is eliminated. During the object tracking period, we also added the Kalman filter algorithm to estimate the position of zebrafish in each frame based on past positions, thereby reduced false positives (FPs). All the functions designed in this study are mainly based on Python and the OpenCV package, making setup adaptation versatile and updates more convenient and efficient.

To evaluate the reliability and accuracy of this setup, acute alcohol exposure experiments with different concentrations of alcohol was performed. We used the established zebrafish detection and behavior assay method to analyze the behavioral parameters of fish such as speed, distance, erratic times, escape times, and motion state. We also conducted a comprehensive evaluation of the setup, by evaluating the accuracy of object detection (precision, recall, F-score) and using the XGBoost to predict alcohol concentration, to verify our behavior evaluation method.

## Materials and methods

### Zebrafish housing

Eighty-four zebrafish (one-month-old) of AB type, provided by Nanjing EzeRinka Biotechnology Co., Ltd, were used in the current study. The purchased fish were placed in the laboratory for 7 D to acclimatize laboratory conditions before the experiment and housed under a photoperiod (14 h light: 10 h dark) throughout the study. During this period, juvenile shrimps were

fed twice a day in the morning and evening. Zebrafish were maintained in 4 L tanks (20–30 fish per tank) with system water (dechlorinated tap water) at 28±2˚C, pH 7.2–7.5 and conductivity 700 muS. The water in the tank is filtered through the sponge and activated carbon particles in the recycler. All fish used in this experiment were untested, healthy, and without any disease. The zebrafish experiments were approved by the ethics committee of the Institutional Animal Care and Use Committee (IACUC) of the Nanjing University of Science and Technology.

## Alcohol effects on zebrafish behavior

Eighty-four zebrafish aged 1 month were first used to determine the optimal alcohol concentration, where zebrafish had the most active behavior and little harm with the mortality rate under 5.0%. Male and female pooled zebrafish were randomly assigned to seven groups exposed to alcohol with concentrations ranging from 0.0% to 3.0% (concentration step: 0.5%). Five minutes after exposure, zebrafish movements were recorded and behavioral characteristics were analyzed. The complete experimental process is shown in Fig 1A.

## Behavioral experiments

Behavior experiment was performed as shown in Fig 1B. After 5 min accustomization, zebrafish in each group were exposed to alcohol with concentrations ranging from 0.0% to 3.0% (concentration step: 0.5%) for 20 min and their behavior were video captured. The research algorithms for video data acquisition and analysis were written in python. Due to the low tracking flux of single target, the most appropriate method is the tracking of multiple objects simultaneously. Zebrafish behavior data were analyzed and stored as Excel files. We use PyCharm edition (an open-source python editor) version 2021.1 to write scripts for tracking and automatic analysis of zebrafish movement characteristics. All codes are detailed in the S1 File.

**Behavioral operating box and video recording system.** The core of this novel video-recording system is composed of a credit card size single-board computer (SBC), namely Raspberry Pi 4 (http://raspberrypi.org), and a Raspberry Pi HQ Camera. In addition, because of the small storage space of Raspberry Pi, the external mobile hard disk is used to store the recorded video files. Monitor and keyboard are optional. To reduce environmental and human interference, we customize a closed operant box made of acrylic (35cm*40cm*60cm, L*W*H), of which the four sidewalls are opaque, and there was a hole with a diameter of 5 cm at the top to fix Raspberry Pi. Furthermore, a groove was arranged at the bottom to insert the LED plate, and a side door was set to put into the dish containing zebrafish. The multi-well plate is placed directly under our video recording system, and the white LED panel is placed under it to enhance the contrast of zebrafish to the surrounding water environment, thus improving zebrafish imaging. The operant box and video recording system mutually form a complete and isolated behavioral assay system (Fig 1B) that does not cause pressure on zebrafish, alleviating unnecessary environmental interference on the behavior.

**Video analysis.** Video tracking is the basis for other behavioral experiments. Only by establishing a system that can automatically obtain the trajectory of zebrafish in the video, can subsequent behavioral analyses be carried out. We designed a video analysis system to accurately identify and track zebrafish, and automatically record information. Subsequently, we define a variety of behavior endpoints to analyze zebrafish behavior. Finally, we automatically calculate the motion behavior parameters of each zebrafish. The complete video analysis process can be divided into the following three stages: image processing (preprocessing and

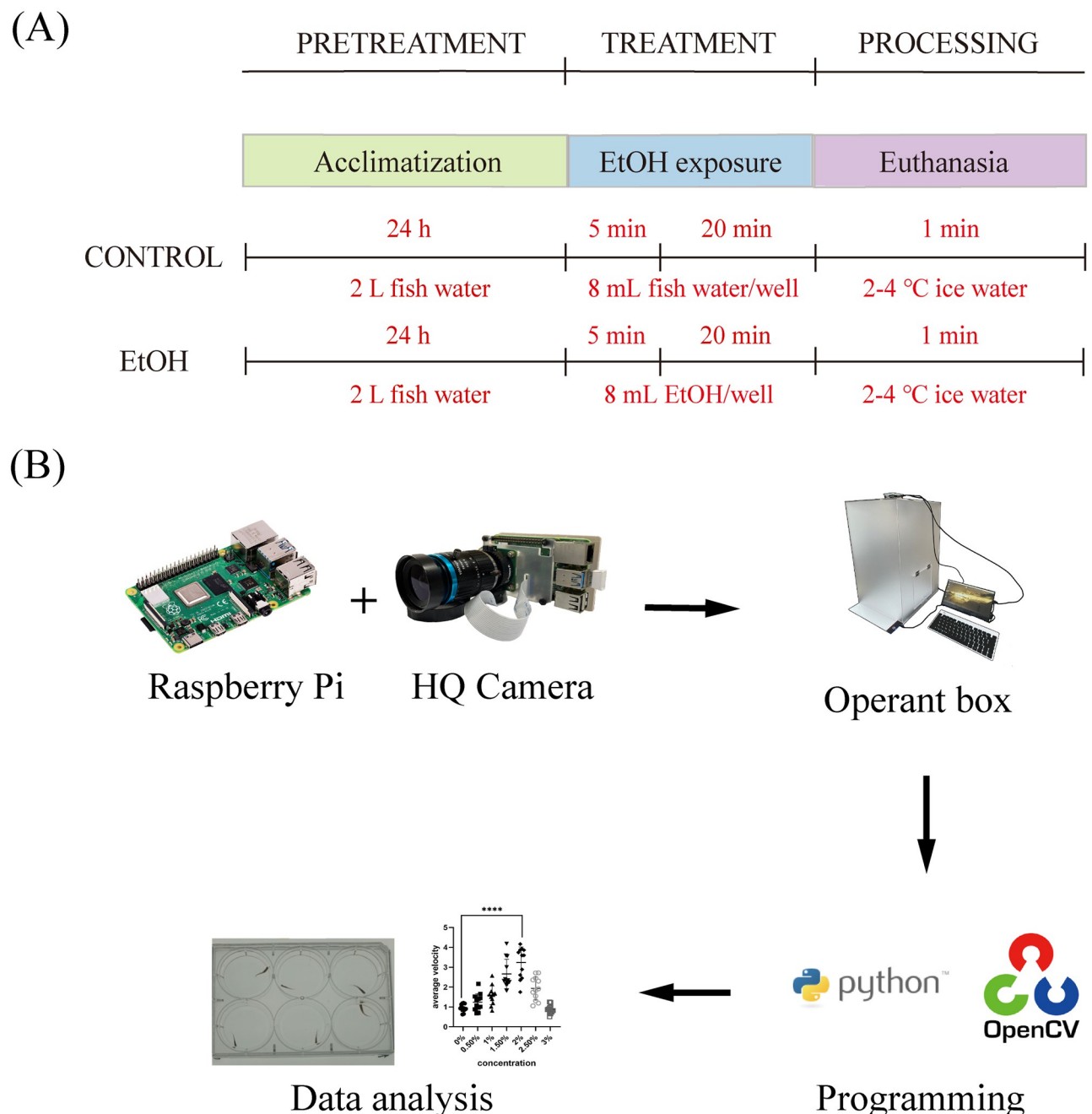

**Fig 1. Methodological diagram for assessing the behavior of zebrafish exposed to alcohol.** (A) Zebrafish were firstly adapted to the environment for 24 h, and then exposed to different concentrations of alcohol for 20 min for behavioral assessment, and finally euthanized in ice water. (B) Flow chart of behavior analysis depicting the composition of the video acquisition system, the operant box, and programming and data processing units.

post-processing) and object recognition; object matching for multiple zebrafish in adjacent frames; continuous tracking of multiple objects between adjacent frames.

**Image processing.** Because the background in the video sequences is relatively stable, that is, each frame has the same background picture, while the zebrafish in the picture are moving and considered as foreground. Since that the camera is fixed, pixels that maintain the same image position in a continuous frame are considered as background pixels. For a video

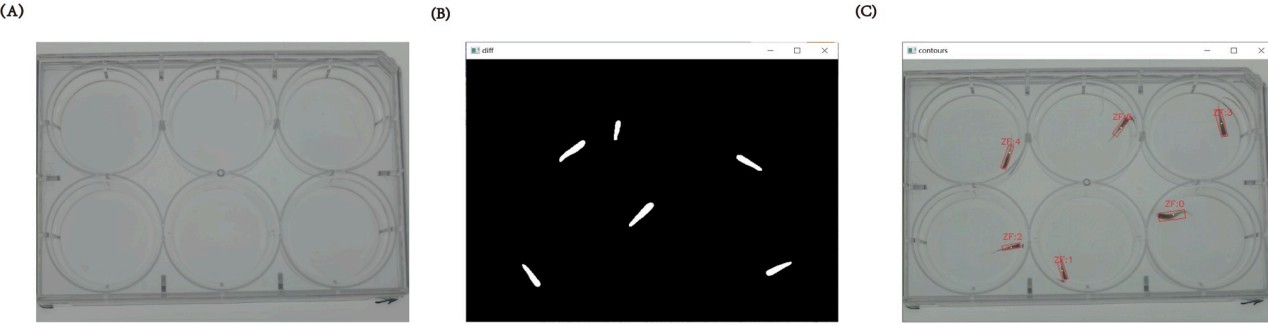

**Fig 2. Image from video analysis.** (A) Video background image. (B) Binary images (C) Object matching results for each zebrafish in continuous frame images.

sequence, we can randomly select $Th_{sp}$ frame images for median background estimation through random sampling. That is, for each pixel, we have $Th_{sp}$ background estimations. As long as a pixel is not covered by zebrafish for more than 50% of the time, the median of the pixel in these $Th_{sp}$ frames will provide a good estimation for the background of the pixel. We use a background extractor to operate on each pixel in the image and then get the background image of the video, as shown in Fig 2A.

In the image pre-processing section, we subtract the current frame and the extracted background image to find the maximum pixel value of each channel. Then the image pixel is valued in the range of [0,255], and the gray image is obtained. To determine the movement range of each zebrafish, regions of interest (ROIs) were defined to eliminate the false recognition caused by the mirror effect (fish reflection on the well-wall) of zebrafish in the multi-well plate and the possible influence of the edge of the multi-well plate. To effectively identify zebrafish in the multi-well plate, we need to convert the gray image into a binary image, as shown in Fig 2B. Different from the simple threshold (the whole image using the same threshold) and adaptive threshold (different regions of the same image using different thresholds), here we use Otsu's binarization method to get the binary image [18]. Firstly, we use median blur to remove noise, and then convert the image into a bimodal image (two peaks in the image histogram). Then, a value between the two peaks is calculated automatically as a threshold to separate them and minimize the variance within each peak. In the image post-processing section, we find the object contours on the binary image obtained by the pre-processing. Here, the minimum threshold $Th_{min}$ and the maximum threshold $Th_{max}$ are set as the area interval, to find the zebrafish. All thresholds used are shown in Table 1.

**Object matching.** We register the zebrafish identified in the first frame as ZF0 ~ ZF5 and store the center and box points (four points in the minimum rectangle surrounding the object) of each zebrafish in each frame. To match the multiple zebrafish identified by the current frame with the previous frame, two conditions need to be met. Firstly, we need to calculate the distance between the centroids of the adjacent frames. The minimum one is considered to belong to the same object and assigned to the same ZF. To increase the accuracy of object

**Table 1. Image processing thresholds.**

| Threshold | Value | Description |
|-----------|-------|-------------|
| $Th_{sp}$ | 160 | Total frames for background extraction |
| $Th_{min}$ | 60 | Minimum object area |
| $Th_{max}$ | 800 | Maximum object area |

matching, we also need to determine whether the centroids allocated under the same ZF are in the same ROI. If both conditions are met at the same time, the matched centroid coordinates (x(t), y(t)) and box points are recorded for subsequent analysis of zebrafish behavior end-points. Repeating the above operations on multiple identified zebrafish finally get information on all objects in the current frame and complete object matching, as shown in Fig 2C.

**Object tracking.** When zebrafish move close to the wall, they may disappear in some frames. To track the movement of fish accurately in a long time and minimize the information loss of each fish in a large number of continuous frames of the video, it is necessary to apply some more complex algorithms. Kalman filter is an efficient autoregressive filter and a power-ful and versatile tool to estimate the internal state of a dynamic system with many uncertainties from combined information [19]. The motion of zebrafish between adjacent frames is consid-ered to be uniform and linear, so the system can be estimated as a dynamic system, within the scope of Kalman filter for tracking task. This algorithm first determines the state vector of fish centroid and box points. The bounding box of fish detected in a frame can be expressed as (x, y, a, h), where (x, y) is center, a is the aspect ratio, and h is the height. Thus, the state vector $x_k$ is defined as $[c_x, c_y, a, h, v_{cx}, v_{cy}, v_a, v_h]^T$. The state vector $x_k$ and measurement equation $z_k$ of the Kalman filter can be expressed as:

$$\begin{cases} x_k = Ax_{k-1} + Bu_k + w_k \\ z_k = Hx_k + v_k \end{cases} \tag{1}$$

where A, B, and H are the state transition matrix, control matrix, and measurement matrix at time k, respectively. Both $w_k$ and $v_k$ are noise conforming to Gaussian distribution, as shown in Eq (2).

$$\begin{cases} w_k \sim N(0, Q_k) \\ v_k \sim N(0, R_k) \end{cases} \tag{2}$$

The first step of the Kalman filter is to initialize the measurement vector $z_k$ and to create a track from unassociated measurement, to obtain the state vector $x_k$ of the first frame and the posterior error covariance matrix $P_k$ (the error covariance matrix). The second step is to pre-dict the state vector $\hat{x}_k$ and its error covariance matrix $\hat{P}_k$ of the current frame based on the previous frame, which can be described as:

$$\begin{cases} \hat{x}_{k|k-1} = Ax_{k-1|k-1} + Bu_k + w_k \\ \hat{P}_{k|k-1} = AP_{k-1|k-1}A^T + Q \end{cases} \tag{3}$$

$$A = \begin{bmatrix} 1 & 0 & 0 & 0 & dt & 0 & 0 & 0 \\ 0 & 1 & 0 & 0 & 0 & dt & 0 & 0 \\ 0 & 0 & 1 & 0 & 0 & 0 & dt & 0 \\ 0 & 0 & 0 & 1 & 0 & 0 & 0 & dt \\ 0 & 0 & 0 & 0 & 1 & 0 & 0 & 0 \\ 0 & 0 & 0 & 0 & 0 & 1 & 0 & 0 \\ 0 & 0 & 0 & 0 & 0 & 0 & 1 & 0 \\ 0 & 0 & 0 & 0 & 0 & 0 & 0 & 1 \end{bmatrix} \tag{4}$$

where Q is the noise state covariance matrix, $P_{k-1}$ is the error covariance matrix at k-1, dt

means the time interval of adjacent frames. When the predicted state vector $\hat{x}_k$, error covariance matrix $\hat{P}_k$, and measurement vector $z_k$ are obtained, the measurement-corrected state vector $x_k$ and the error covariance matrix $P_k$ at time k can be calculated by Eq (5). I, H, R, K represent the unit matrix, measurement matrix, the observed noise covariance matrix, and the Kalman gain, respectively, as shown in Eq (7).

$$\begin{cases} x_{k|k} = \hat{x}_{k|k-1} + K(z_k - Hx_{k|k-1}) \\ P_{k|k} = (I - KH)\hat{P}_{k|k-1} \end{cases} \tag{5}$$

$$H = \begin{bmatrix} 1 & 0 & 0 & 0 & 0 & 0 & 0 & 0 \\ 0 & 1 & 0 & 0 & 0 & 0 & 0 & 0 \\ 0 & 0 & 1 & 0 & 0 & 0 & 0 & 0 \\ 0 & 0 & 0 & 1 & 0 & 0 & 0 & 0 \end{bmatrix} \tag{6}$$

$$K = \hat{P}_{k|k-1}H^T(HP_{k|k-1}H^T + R)^{-1} \tag{7}$$

Finally, we input the corrected k-time vector as the state vector of Kalman in the next frame and record the information (centroids and box points) for subsequent zebrafish behavior analysis.

**Behavior parameters.** To quantify the behavioral characteristics of zebrafish, we developed a set of parameters to describe their behavior in various situations, including distance, velocity, maximum velocity, angle, angular velocity, thigmotaxis, meander, erratic motions, escape times, and left/right deflection times. The mobility state was described based on zebrafish speed to distinguish freezing, swimming, and rapid duration times or percentage. The definition of specific parameters is shown in Table 2.

## Detection and tracking evaluation

Seven video sequences are randomly selected from twenty videos of two experiments to evaluate the performance of the object recognition and tracking algorithm. Details of these video sequences are shown in Table 3. Since the number of zebrafish in the video is known, the performance of the algorithm is judged by recording whether each frame of zebrafish is correctly identified. The validation results are assessed using the standard measures of precision (P), recall (R) and F-score (F) as defined below.

$$Precision(P) = \frac{TP}{TP + FP}, \; Recall(R) = \frac{TP}{TP + FN}, \; F - score(F) = \frac{2PR}{(P + R)}$$

## Model accuracy evaluation

To evaluate the accuracy of the zebrafish behavior assay method established in this study, the dataset was divided into a training set and a test set in a ratio of 5:1 and subjected to fivefold cross-validations using the XGBoost method to predict alcohol concentration and judge the prediction accuracy.

**Table 2. Definition of behavioral endpoints.**

| Parameters | Formula | Description |
|---|---|---|
| Distance (cm) | $\Delta s_i = \sqrt{(x_i - x_{i-1})^2 + (y_i - y_{i-1})^2}$ <br><br> $s = \sum_{i=1}^{t} \Delta s_i$ | The total distance of zebrafish swimming within a certain time, namely the sum of the Euclidean distance of the zebrafish position in all consecutive frames. |
| Velocity (cm/s) | $v_i(t) = \left. {}^{d\Delta s_i} \middle/ {}_{dt} \right., i \in [1, n-1]$ | Distance of zebrafish swimming per unit time. |
| Maximum velocity (cm/s) | $Max\{v_1, \ldots, v_i, \ldots, v_{n-1}\}$ | A maximum speed of fish swimming within 20 min. |
| Angle (deg) | $ang = \arccos\left(\frac{dot(vect_1, vect_2)}{\|vect_1\|\|vect_2\|}\right) \times \frac{180}{\pi}$ | The rotation angle of zebrafish between consecutive frames. |
| thigmotaxis | $thig = \frac{time(wall)}{time(wall) + time(center)}$ | The time in the outer circle versus the total times. |
| Parameters | Formula | Description |
| Meander (deg/cm) | $meander = {}^{ang} \middle/ {}_{\Delta s_i}$ | The sum of rotation angles of zebrafish within unit distance. |
| Erratic movements | $v_i(t) > 3, erratic+ = 1$ | Times when the velocity change of zebrafish is greater than 3 cm/s. |
| Escape times | $\begin{cases} v_i(t) > 5 \\ ang > 45° \end{cases}, escape+ = 1$ | Times when the velocity and angle of zebrafish change greatly at the same time. |
| Freezing duration time (s) | $0 < v_i(t) < 1, times+ = 1$ <br><br> $duration\_time = {}^{times} \middle/ {}_{fps}$ | Duration time when the velocity of zebrafish is less than 1 cm/s. |
| Freezing time percentage (%) | $0 < v_i(t) < 1, times+ = 1$ <br><br> $percentage = {}^{times} \middle/ {}_{n}$ | Percentage of the state when thevelocity of zebrafish is less than 1 cm/s. |
| Swimming duration time (s) | $1 < v_i(t) < 10, times+ = 1$ <br><br> $duration\_time = {}^{times} \middle/ {}_{fps}$ | Duration time when the velocity of zebrafish is between 1–10 cm/s. |
| Parameters | Formula | Description |
| Swimming time percentage (%) | $1 < v_i(t) < 10, times+ = 1$ <br><br> $percentage = {}^{times} \middle/ {}_{n}$ | Percentage of the state when the velocity of zebrafish is between 1–10 cm/s. |
| Rapid duration time (s) | $v_i(t) > 10, times+ = 1$ <br><br> $duration\_time = {}^{times} \middle/ {}_{fps}$ | Duration time when the velocity of zebrafish is large than 10 cm/s. |
| Rapid time percentage (%) | $v_i(t) > 10, times+ = 1$ <br><br> $percentage = {}^{times} \middle/ {}_{n}$ | Percentage of the state when the velocity of zebrafish is large than 10 cm/s. |
| Left/Right times | $det = det(vect_1, vect_2)$ <br><br> $\begin{cases} det < 0, right+ = 1 \\ det > 0, left+ = 1 \end{cases}$ | Times of left and right deflections in the process of zebrafish moving. |

**Table 3. Automatic zebrafish recognition and tracking assessment.**

| Video dataset | Fish number | Truth | TP | FP | FN | P | R | F |
|---|---|---|---|---|---|---|---|---|
| D1 | 6 | 29959 | 29878 | 14 | 67 | 0.9995 | 0.9978 | 0.9986 |
| D2 | 6 | 29958 | 29914 | 24 | 20 | 0.9992 | 0.9993 | 0.9993 |
| D3 | 6 | 29955 | 29955 | 0 | 0 | 1 | 1 | 1 |
| D4 | 6 | 29955 | 29914 | 0 | 41 | 1 | 0.9986 | 0.9993 |
| D5 | 6 | 29954 | 28956 | 20 | 978 | 0.99931 | 0.967328 | 0.983059 |
| D6 | 6 | 29955 | 29644 | 42 | 269 | 0.998585 | 0.991007 | 0.994782 |
| D7 | 6 | 29955 | 29872 | 42 | 41 | 0.998596 | 0.998629 | 0.998613 |

TP, true positive; FP, false positive; FN, false negative; P, precision; R, recall; F, F-score.

### Statistical analysis

In this study, generalized linear model (GzLM) was used to evaluate the influence of alcohol concentration on behavioral parameters. Both R and Python were used for statistical analysis and plotting.

## Results

### Automatic zebrafish detection and tracking

To assess the zebrafish detection and tracking algorithm, we analyzed the data of seven randomly selected video sequences and recorded times that zebrafish correctly detected in each video sequence. True positive (TP) and false negative (FN) means success and failure in detection of zebrafish in the current frame, respectively. Error detection of other objects as zebrafish due to water droplets, impurities, reflections, and other interference is considered to be false positive (FP). We found that the range of ground truth data is between 29950 and 29960. It can be observed from Table 3 that our proposed algorithm presents a superior performance with high accuracy and recall. It is worth noting that the precision, recall, and F-scored of the D3 dataset reached 1.

### Alcohol affects zebrafish behavior differently according to the concentration

To verify the feasibility of the automatic zebrafish behavioral analysis system proposed in this study, the experiment was designed for comparative verification.

Using the zebrafish behavior analysis system proposed in this study, we quantified the movement behavior of zebrafish and evaluated a variety of behavioral parameters after exposure to different concentration of alcohol. These data were supported by the representative tracking, three-dimensional reconstruction and occupancy plots that illustrate the traits and the occurrence frequency of zebrafish in different concentrations of EtOH. The reddish region in occupancy plots indicated more relatively frequent occurrence of zebrafish. Zebrafish in the control group (0.0%) had relatively uniform movement, mainly low-speed movement. With the increase of alcohol concentration (0.5–2.0%), zebrafish movements increased rapidly, and peaked at 2.0%, which was manifested by the increase of red and yellow area in the occupancy plot (Fig 3). Under high concentrations (2.5–3.0%) of alcohol exposure, the tracking plot and three-dimensional reconstruction (Fig 3) showed that zebrafish activity waned towards freezing states, which was characterized by reduced trajectory and slower velocity.

Different concentrations of alcohol had various affluence on the behavior of zebrafish. As shown in Fig 4A, the distance, velocity and maximum velocity of zebrafish gradually increased at low concentrations of alcohol and reached the maximum at 2.0% concentration. With the continuous increase of alcohol concentration, the distance and speed of zebrafish decreased significantly, which were equivalent to those in the control group. In addition, the angle, angular velocity, meander, and other behavioral parameters also denoted that the movement of zebrafish is most dynamic at 2.0% alcohol concentration. At 2.0% ethanol concentration, escape time and erratic movements had the highest values, while zebrafish in different groups had no significant difference in swimming direction preference. We also evaluated the mobility (freezing, swimming, and rapid movement) of zebrafish, as shown in Fig 4B. We can find that the control group (0.0%) has the highest proportion of freezing state, while under low concentration (0.5–1.0%) alcohol exposure, the swimming and freezing state of zebrafish accounts for about 40% and 60%, respectively, and there has almost no rapid state. With the increase of alcohol concentration (1.5–2.5%), the swimming state of zebrafish gradually increased to

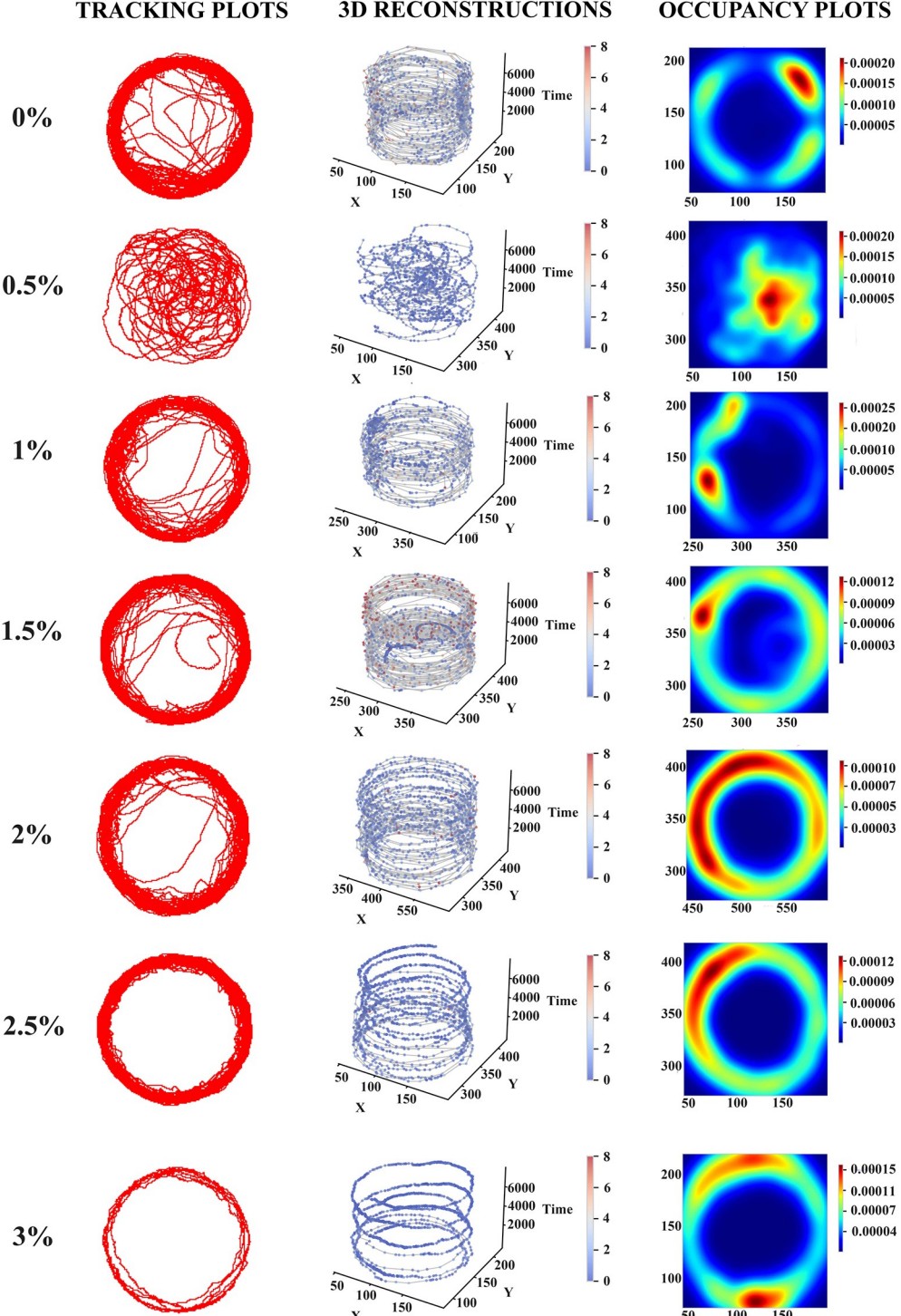

**Fig 3. Effect of different concentrations (0.0%, 0.5%, 1.0%, 1.5%, 2.0%, 2.5%, 3.0%) of EtOH treatment on the mobility of zebrafish.** Tracking plot represents the motion trajectory of zebrafish. Three-dimensional reconstruction adds time series (Z-axis) based on two-dimensional trajectory (X- and Y-axis). The velocity of the experiment is expressed by a color scale gradient, from low velocity (dark blue) to high velocity (dark red). Occupancy plot is obtained according to the frequency of zebrafish occurrences. Trajectory and occupancy plot are from a single fish. The data obtained by our proposed behavioral analysis system display the overall behavior of each experimental group during the last 5 min trail.

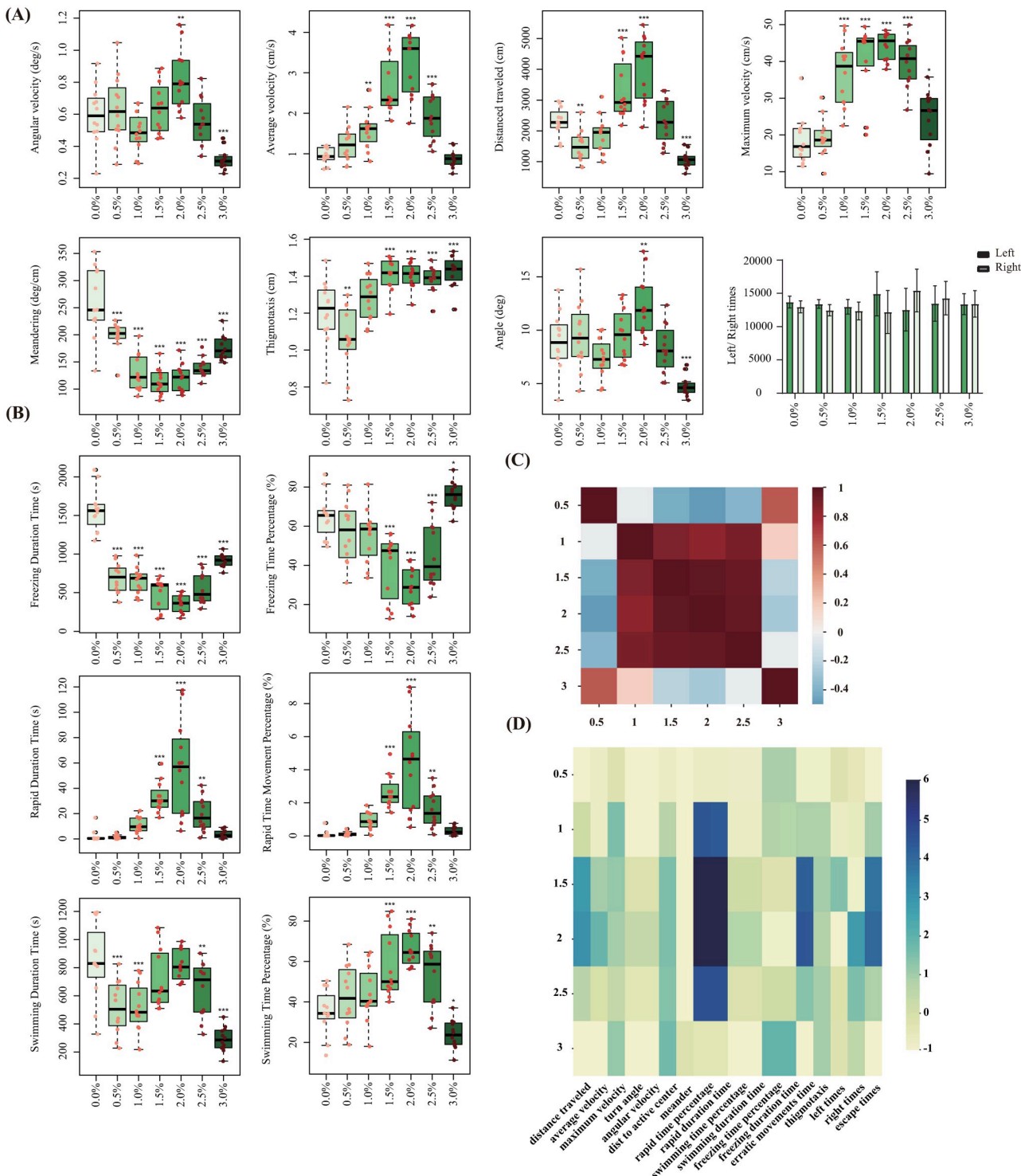

**Fig 4. Behavioral changes of zebrafish exposed to acute alcohol at different concentrations (0.0%, 0.5%, 1.0%, 1.5%, 2.0%, 2.5%, 3.0%).** (A) The behavior endpoints of zebrafish within 20 minutes, including angular velocity, swimming speed, angle, distance, maximum speed, etc. (B) The duration and percentage of the three states (freezing, swimming, fast movement) of zebrafish. (C) Heat map of correlation coefficients among different groups. (D) Eigenvector Matrix map of different alcohol concentrations. All data were obtained by the generalized linear model (GzLM). (*p< 0.05, **p< 0.01, ***p< 0.001 and ****p< 0.0001. *Significant difference from 0% group).

about 60%, and the freezing state decreased to 35%. In high concentration (3.0%) alcohol exposure, the freezing state reached 80%, even higher than the control group (0.0%).

After obtaining the behavior parameters, in order to analyze the influence of the experimental group on the behavior of zebrafish, we generated the feature vector describing the behavior of zebrafish in the experimental group compared with that in the control group. To compare the resemblance between different groups of zebrafish behavior, we calculated the uncentered Pearson correlation coefficient, as shown in Fig 4C. It can be seen that the behavior of zebrafish is similar under 0.5% and 3.0% alcohol concentration. The feature vectors at different concentrations constitute a matrix, which can be visualized as a heat map to intuitively reflect the effect of alcohol, as shown in Fig 4D.

### Prediction of alcohol concentration using XGBoost

Based on the results of experiment (Alcohol effects on zebrafish behavior), the behavior endpoints of zebrafish were used as the predictor to predict alcohol concentration the zebrafish exposed. The dataset was divided into the training set and test set for fivefold cross-validation using XGBoost, which give the importance of each feature, and the accuracy of the alcohol concentration prediction model. The prediction results may be different due to the randomness of the program. Therefore, the average accuracy of fivefold is considered as a criterion, which can explain the deviation of the alcohol prediction model. As shown in Fig 5, the bar graph of feature importance is obtained. It can be intuitively seen that the importance of the five features (freezing duration time, meander, rapid duration time, distance, and velocity) are relatively higher. The accuracy of alcohol concentration prediction ranges between 75.0% and 83.33%, as shown in Fig 6.

### Discussion

In this study, we built a high-throughput zebrafish behavior analysis setup based on Raspberry Pi to analyze various motion behaviors and motion states of zebrafish. Behavior endpoints of zebrafish are shown in Table 2. Firstly, video data is obtained through HQ Camera and Raspberry Pi of this behavior analysis setup, and then automatically analyzed for behavior parameters according to the proposed algorithm. We evaluated the performance of the zebrafish detection and tracking algorithm and tested the performance of the setup proposed in this study: different concentration of alcohol exposure experiment. Finally, the accuracy of this algorithm was fully assessed by predicting the alcohol concentration through the XGBoost method.

The precision and recall of the automatic zebrafish detection and tracking algorithm proposed in this study are satisfactorily high with good performance. During the process of video analysis, we try to reduce the number of missing zebrafish (FP), so as to improve the precision of zebrafish detection. At the same time, the false detection object (FN) is reduced, that is, the factors such as reflection and water droplets are excluded to improve recall. The high accuracy of the object detection and tracking algorithm ensures the reliability of the subsequent behavior analysis algorithm. Acute alcohol exposure demonstrates that the behavior analysis algorithm can accurately capture the behavior of zebrafish.

Acute alcohol exposure can induce typical behavioral changes that can be measured in simple, rapid tasks [20]. In this study, we analyzed the data at the last 5 min of 20 min zebrafish alcohol exposure. The changes in the overall behavior of zebrafish were evaluated according to the tracking, three-dimensional reconstruction and occupancy plot. We also observed significant changes in some behavioral activities of zebrafish in the video of 20 min acute alcohol exposure, such as average speed, maximum speed, swimming distance, and meander.

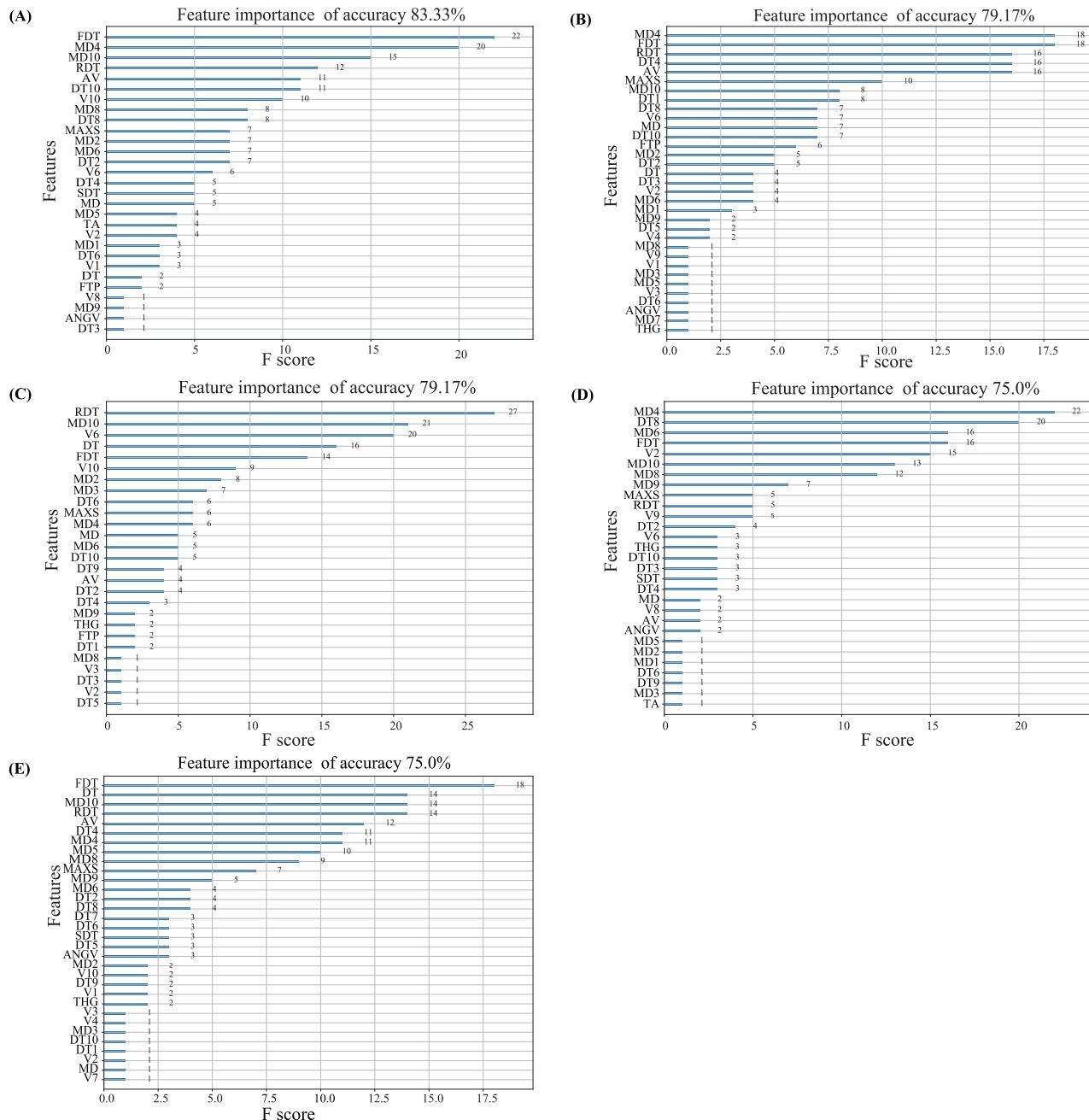

**Fig 5. Feature importance bar graph for five cross-validations.** FDT: freezing duration time, MD: meandering, RDT: rapid duration time, AV: average velocity, DT: distance traveled, V: velocity, MAXS: maximum speed, SDT: swimming duration time, TA: turn angle, FTP: freezing time percentage, ANGV: angular velocity, THG: thigmotaxis.

Moreover, these behavioral parameters showed an obvious inverted U-shaped phenomenon, that is, low concentration of alcohol exposure (0.5–2.0%) promoted zebrafish movement, and high concentration (2.5–3.0%) inhibited movement. These results indicated that the low concentration of acute alcohol exposure could reduce the anxiety of zebrafish, while the high concentration of alcohol played a sedative role, making the behavior of zebrafish slow, which

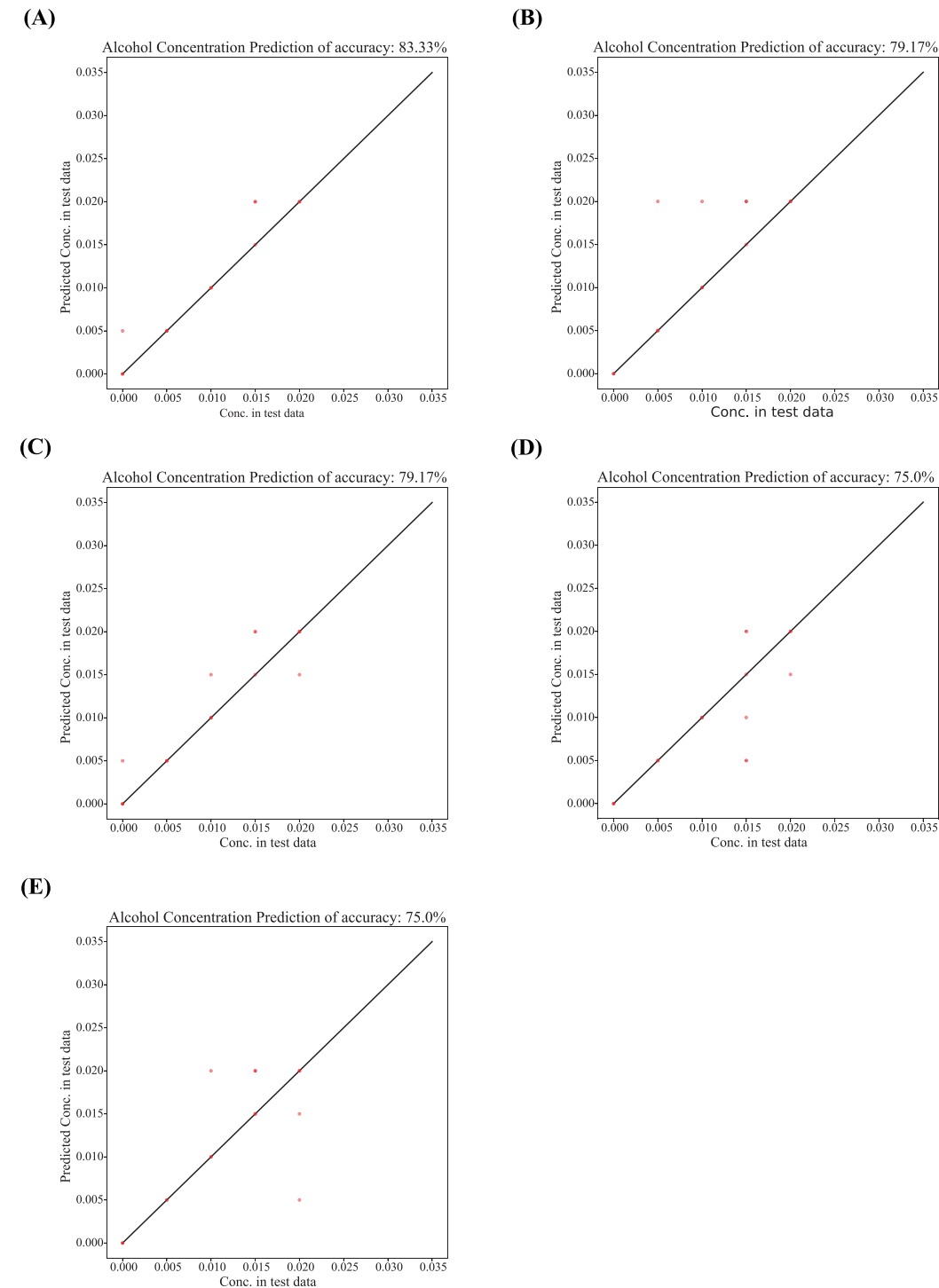

**Fig 6. Accuracy evaluation of alcohol concentration prediction.**

might be caused by alcohol inhibition on nerve system of zebrafish. Our result is consistent with previous study on alcohol exposure effects on in zebrafish behavior [21, 22]. This phenomenon is also reflected in the motion state of zebrafish. The control group (0.0% EtOH) was mainly freezing, indicating that the fish of the control group are cautious to the new

environment. With the increase of alcohol concentration (0.5–2.0%), the freezing state gradually decreased and the swimming state increased, indicating that the caution state of zebrafish was weakened and its behavior became more excited. However, it is worth noting that fish under high concentrations of alcohol (2.5–3.0%) are mostly in a state of freezing, which may be due to their inhibited autonomous movement by alcohol.

In addition, we used the XGBoost method to predict alcohol concentration based on the data of acute alcohol experiment (Alcohol effects on zebrafish behavior) to evaluate the accuracy of our proposed behavior analysis algorithm. The results show that the prediction accuracy is high, all above 75.0% (Figs 5 and 6), that is, the algorithm proposed in this study performs well. We observed the importance of freezing duration time, meander, fast duration time, distance, speed these five features are relatively high, indicating that these five kinds of behavior changes are most obvious in zebrafish. In addition, we also find that the performance of the model, namely accuracy, usually decreases with the number of selected features, so it is necessary to balance the number of features. Our automatic detection and behavior analysis system ensures the accuracy of the behavior data. In this study, higher frame frequency video will not produce better results, but increase the algorithm execution time and reduce efficiency. Therefore, we set the frame frequency of the video to 25 fps. Alcohol concentration experiment (Alcohol effects on zebrafish behavior) proved that the proposed behavioral analysis method can capture the U-shaped dose-dependent phenomenon in zebrafish, which was consistent with the reported results. Therefore, this method proposed in this study can accurately detect the behavioral activities of zebrafish, and evaluate the drug toxicity or drug treatment effect through the behavioral changes of zebrafish.

Although some commercial systems have been able to implement these experiments, the proposed behavior analysis method does not require additional costs. In this study, we built a reliable, low-cost device to demonstrate that zebrafish behavioral endpoints can be used as alternative biomarkers for screening potential drugs and evaluating drug effects. More importantly, a large number of experiments can be carried out simultaneously, and the behavior parameters will be obtained after capturing the video data. Therefore, large-scale, high-throughput studies can be implemented. We hope that this analysis system can help reduce the subjective bias of the experimenter, reduce the experimental operation error, and analyze the experimental results more intuitively and quickly in the behavioral analysis of zebrafish. In the future, we believe that this approach can be used to explain the behavior of zebrafish in a variety of neurological disease models, screening drugs, and other studies.

## Conclusion

In this paper, a novel setup was designed, which can be used to obtain video data and determine a variety of functional parameters to reliably evaluate the behavioral performance of zebrafish. The reliability of this method was verified by alcohol concentration prediction, and the accuracy of this method was evaluated by the XGBoost prediction of alcohol concentration. We believe that the zebrafish behavior analysis setup proposed in this study can be easily accustomed to be used in drug screening, efficacy and toxicity evaluation, disease behavior assessment, and other fields.

## Supporting information

**S1 File. Codes.**
(RAR)

**S2 File. Data.**
(RAR)

**S3 File. Videos.**
(RAR)

**S1 Fig. Parameter QQplot.**
(EPS)

**S1 Graphical abstract.**
(TIF)

## Author Contributions

**Data curation:** Yunlin Li, Benxing Yao, Ran Xiao, Yudie Hu.

**Formal analysis:** Wenya Liu, Guanghui Li.

**Investigation:** Fengye Wu, Qinyan Wu.

**Methodology:** Yunlin Li.

**Software:** Yunlin Li.

**Supervision:** Junsong Wang.

**Validation:** Yunlin Li.

**Writing – original draft:** Yunlin Li.

**Writing – review & editing:** Junsong Wang.

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
