## [Decision Letter · Decision Letter 0]

6 Nov 2022

PONE-D-22-20565A novel open-source raspberry Pi-based behavioral testing in zebrafishPLOS ONE

Dear Dr. Wang,

Thank you for submitting your manuscript to PLOS ONE. After careful consideration, we feel that it has merit but does not fully meet PLOS ONE’s publication criteria as it currently stands. Therefore, we invite you to submit a revised version of the manuscript that addresses the points raised during the review process.

We look forward to receiving your revised manuscript.

Kind regards,

Hai O. Xu

Academic Editor

PLOS ONE

Journal Requirements:

Reviewers' comments:

Reviewer's Responses to Questions

**Comments to the Author**

1. Is the manuscript technically sound, and do the data support the conclusions?

Reviewer #1: Partly

Reviewer #2: Yes

2. Has the statistical analysis been performed appropriately and rigorously? 

Reviewer #1: Yes

Reviewer #2: No

3. Have the authors made all data underlying the findings in their manuscript fully available?

Reviewer #1: No

Reviewer #2: No

4. Is the manuscript presented in an intelligible fashion and written in standard English?

Reviewer #1: Yes

Reviewer #2: No

5. Review Comments to the Author

Reviewer #1: Thank you for sharing your manuscript with me. The author designed a low-cost automatic zebrafish behavior detection device and software to measure zebrafish behavior parameters. The results show that the device designed by the author can be used to obtain video data and determine various behavioral parameters to reliably evaluate the behavioral performance of zebrafish. Generally speaking, the theme of the manuscript is very meaningful and necessary. But there are many problems that need to be corrected. These issues are listed below and may help improve the manuscript.

1. Figure 4 is not clear enough. The horizontal and vertical coordinates of each result cannot be seen clearly. Please replace the result map with a higher definition.

2. What is the composition of the system water mentioned in line 91?

3. The zebrafish of one month old is used in the experiment. Can you track and analyze the behavior of small fish several days old after fertilization or adults of three months old through this device and software?

4. Line 163: regions of interest (ROIs) were defined to eliminate the false recognition caused by the mirror effect (fish reflection on the well-wall) of zebrafish in the multi-well plate and the possible influence of the edge of the multi-well plate. The region of interest here refers to which region?

Reviewer #2: This study provides a system package for determining multiple behavioral parameters of zebrafish, a model organism in the behavioral tests. In addition, the authors used an alcohol exposure test to examine the validity of their method. The MS has some novelty, but the following issues must be clarified before being recommended for publication.

（1）Tables: Three-line tables should be used, and footnotes should be used to indicate the meaning of abbreviations.

（2）Statistical analysis method: Since behavioral data usually cannot satisfy normal distribution and Homogeneity, one-way ANOVA and unpaired t-test are not necessarily applicable. At this time, it is best to use the generalized linear model (GzLM) for analysis. See the references:

Persistent impact of amitriptyline on the behavior, brain neurotransmitter, and transcriptional profile of zebrafish (Danio rerio). Aquatic Toxicology, 2022, 245 10.1016/j.aquatox.2022.106129.

Impacts of chronic exposure to sublethal diazepam on behavioral traits of female and male zebrafish (Danio rerio). Ecotoxicology and Environmental Safety, 2021, 208 10.1016/j.ecoenv.2020.111747.

The author should provide some videos to give readers an intuitive impression.

The authors should also discuss and identify the innovations and advantages of their methods. For example, using their systems and commercial equipment to track and analyze the trajectory of the same subjects, compare their results, and then identify the advantages and disadvantages.

Social behavior is the most difficult in fish behavioral analysis. Because when multiple individuals are located in the same container, it is easy to cause exchange between subjects (especially when behavioral trajectories cross each other), and it also involves complex interactions between subjects. I would like to know if there is an excellent way to solve the above problem with this methods.

6. PLOS authors have the option to publish the peer review history of their article (what does this mean?). If published, this will include your full peer review and any attached files.

Reviewer #1: No

Reviewer #2: No

---

## [Author Response · Author response to Decision Letter 0]

7 Dec 2022

Response

First of all, we thank the reviewers for your comments and suggestions on our manuscript “A novel open-source raspberry Pi-based behavioral testing in zebrafish”. Based on the constructive and helpful comments, we addressed the raised questions and revised our manuscript carefully. Some changes have been made in this revision and our responses to each point (in blue colors) are as follows:

Reviewer #1: Thank you for sharing your manuscript with me. The author designed a low-cost automatic zebrafish behavior detection device and software to measure zebrafish behavior parameters. The results show that the device designed by the author can be used to obtain video data and determine various behavioral parameters to reliably evaluate the behavioral performance of zebrafish. Generally speaking, the theme of the manuscript is very meaningful and necessary. But there are many problems that need to be corrected. These issues are listed below and may help improve the manuscript.

Q1. Figure 4 is not clear enough. The horizontal and vertical coordinates of each result cannot be seen clearly. Please replace the result map with a higher definition.

Response: Thank you for pointing out the weakness of our paper. According to your suggestion, we have modified the Figure 4 by changing it to a vector image (format: eps or tiff) with higher resolution. In addition, barplots for parameters were replaced by boxplot with individual data points jittered to allow fully and easy comprehension of the data.

Q2. What is the composition of the system water mentioned in line 91?

Response: Sorry for missing of details. This information was added to the paragraph as: “Zebrafish were maintained in 4 L tanks (20-30 fish per tank) with system water (dechlorinated tap water) at 28±2 ℃, pH 7.2-7.5 and conductivity 700 muS.”

Q3. The zebrafish of one month old is used in the experiment. Can you track and analyze the behavior of small fish several days old after fertilization or adults of three months old through this device and software?

Response: Thank you for proposing such an important question. We tracked and analyzed the behavioral videos (around 60 seconds) of 5 days postfertilization (dpf) and adults of three months old zebrafish through our device and software. Both three-month and five-day post-fertilization zebrafish could be identified. The 5 dpf zebrafish had a translucent and very small body, which made identification more difficult and required a higher resolution of the camera. Three-month-old zebrafish are easier to identify because of their relatively larger size and opaque body.

Q4. Line 163: regions of interest (ROIs) were defined to eliminate the false recognition caused by the mirror effect (fish reflection on the well-wall) of zebrafish in the multi-well plate and the possible influence of the edge of the multi-well plate. The region of interest here refers to which region?

Response: The region of interest (ROI) here refers to each well of multi-well plates. We use a python script to mark the position of each well of the multi-well plate on the extracted background image (Fig2. A), and then we use ‘select_circle_roi.py’ in ‘S1 codes/pre-processing’ to further determine the ROI. The image processing effect after adding ROI is shown below. The specific location information of the ROIs is collected once with the same camera and multi-well plate placement.

Reviewer #2: This study provides a system package for determining multiple behavioral parameters of zebrafish, a model organism in the behavioral tests. In addition, the authors used an alcohol exposure test to examine the validity of their method. The MS has some novelty, but the following issues must be clarified before being recommended for publication.

Q1 Tables: Three-line tables should be used, and footnotes should be used to indicate the meaning of abbreviations.

Response: Sorry for making this mistake. Done as suggested.

Q2 Statistical analysis method: Since behavioral data usually cannot satisfy normal distribution and Homogeneity, one-way ANOVA and unpaired t-test are not necessarily applicable. At this time, it is best to use the generalized linear model (GzLM) for analysis. See the references:

Persistent impact of amitriptyline on the behavior, brain neurotransmitter, and transcriptional profile of zebrafish (Danio rerio). Aquatic Toxicology, 2022, 245 10.1016/j.aquatox.2022.106129.

Impacts of chronic exposure to sublethal diazepam on behavioral traits of female and male zebrafish (Danio rerio). Ecotoxicology and Environmental Safety, 2021, 208 10.1016/j.ecoenv.2020.111747.

Response: Thank you for pointing out the weakness of our paper. The normality consumption of these parameters were assessed based on the Shapiro-Wilk normality test and visualized by QQplot. It was found that most of these parameters did not conform to a normal distribution, with p-values less than 0.05. In addition, parameters were not in linear relationship with alcohol concentrations (Figure 4). As suggested, the generalized linear model (GzLM) analysis was performed using a gaussian distribution and identity link function, considering alcohol concentration as categorical variable. Figure 4 in manuscript was thus also reorganized. In addition, barplots for parameters were replaced by boxplot with individual data points jittered to allow fully and easy comprehension of the data.

Q3. The author should provide some videos to give readers an intuitive impression.

Response: Thank you for your valuable comments and advice. According to your suggestion, we have uploaded some videos as supporting information files, including the original video, the zebrafish identification and tracking process video, the mask after ROI operation video and the binarization process video after image processing. In order to comply with the uploaded file size regulations and for quick and easy viewing, only 50 seconds of the experimental video is captured here for reference.

Q4. The authors should also discuss and identify the innovations and advantages of their methods. For example, using their systems and commercial equipment to track and analyze the trajectory of the same subjects, compare their results, and then identify the advantages and disadvantages.

Response: Thank you for proposing such an important question. We used Cohen's Kappa to measure the agreement between the platform proposed in this study and the commercial platform. It was found that the agreement of most of the parameters was greater than 0.33. Among them, the agreement of parameters such as average velocity, rapid duration time and rapid time movement percentage reached 1, which indicates a fair agreement between the two platforms.

Q5. Social behavior is the most difficult in fish behavioral analysis. Because when multiple individuals are located in the same container, it is easy to cause exchange between subjects (especially when behavioral trajectories cross each other), and it also involves complex interactions between subjects. I would like to know if there is an excellent way to solve the above problem with this methods.

Response: Thank you for proposing such an important problem. In this study, a multi-well plate was used as a container to hold multiple zebrafish and ensured only one fish was placed in each well, thus avoiding problems such as trajectories cross each other and targets exchange that may arise from multiple zebrafish in one space. In addition, we further exclude the possibility of target exchanging between zebrafish in adjacent wells due to their close proximity by determining the region of interest (ROI) and using a target matching algorithm.

Specific images and tables are available in 'Response to Reviewers.docx'.

---

## [Editor Report · Decision Letter 1]

12 Dec 2022

A novel open-source raspberry Pi-based behavioral testing in zebrafish

PONE-D-22-20565R1

Dear Dr. Wang,

We’re pleased to inform you that your manuscript has been judged scientifically suitable for publication and will be formally accepted for publication once it meets all outstanding technical requirements.

Kind regards,

Hai O. Xu

Academic Editor

PLOS ONE
---

## [Editor Report · Acceptance letter]

15 Dec 2022

PONE-D-22-20565R1 

A novel open-source raspberry Pi-based behavioral testing in zebrafish 

Dear Dr. Wang:

I'm pleased to inform you that your manuscript has been deemed suitable for publication in PLOS ONE. Congratulations! Your manuscript is now with our production department. 

Kind regards, 

on behalf of

Dr. Hai O. Xu 

Academic Editor

PLOS ONE